# High-*κ* van der Waals Oxide MoO_3_ as Efficient Gate Dielectric for MoS_2_ Field-Effect Transistors

**DOI:** 10.3390/ma15175859

**Published:** 2022-08-25

**Authors:** Junfan Wang, Haojie Lai, Xiaoli Huang, Junjie Liu, Yueheng Lu, Pengyi Liu, Weiguang Xie

**Affiliations:** Siyuan Laboratory, Guangzhou Key Laboratory of Vacuum Coating Technologies and New Energy Materials, Guangdong Provincial Engineering Technology Research Center of Vacuum Coating Technologies and New Energy Materials, Department of Physics, Jinan University, Guangzhou 510632, China

**Keywords:** MoS_2_, MoO_3_, FET, photodetector, field effect transistor, gate dielectric, transistor, van der Waals materials

## Abstract

Two-dimensional van der Waals crystals (2D vdW) are recognized as one of the potential materials to solve the physical limits caused by size scaling. Here, vdW metal oxide MoO_3_ is applied with the gate dielectric in a 2D field-effect transistor (FET). Due to its high dielectric constant and the good response of MoS_2_ to visible light, we obtained a field effect transistor for photodetection. In general, the device exhibits a threshold voltage near 0 V, Ion/Ioff ratio of 10^5^, electron mobility about 85 cm^2^ V^−1^ s^−1^ and a good response to visible light, the responsivity is near 5 A/W at low laser power, which shows that MoO_3_ is a potential material as gate dielectric.

## 1. Introduction

There is no doubt that the transistor is an indispensable part of modern life, it is the brain of all electronic devices today. Unfortunately, the short channel effect caused by the reduction in channel size and the direct electron tunneling caused by the decrease in dielectric layer thickness seriously restricts the performance of devices [1,2,3,4]. Two-dimensional van der Waals materials (2D vdW) with superior carrier migration and excellent gate controllability at atomic thickness are considered as the next-generation potential channel materials [5]. In addition, since the capacitance is inversely proportional to the gate dielectric material thickness and proportional to the dielectric constant [6]. It can be seen that materials with high dielectric constant can satisfy the thickness reduction while providing strong capacitive coupling. To date, the 2D field-effect transistors supported by high-*κ* material substrate has been reported. For example, Ganapathi et al. fabricated MoS_2_ short channel devices with 5 nm-thick HfO_2_, achieving an extremely low threshold voltage (approximately −1 V), 10^6^ switching ratio, and extremely low subthreshold swing (110~120 mV/dec) [7]. Ah-Jin Cho et al. also successfully fabricated the field effect devices using Al_2_O_3_ as the gate dielectric layer [8]. However, there are a lot of dangling bonds and surface trap states in the thin film dielectric layer, resulting in a large hysteresis. High-quality vdW dielectric has received much attention due to its atomic plane, hanging bonds-free, and low trap density. Hexagonal boron nitride (hBN) is the most common two-dimensional insulator and is widely regarded as a promising gate insulator in two-dimensional material transistors. However, the dielectric constant of hBN is only 5 [9]. Recently, Liu et al. showed that MoS_2_ field-effect transistors supported by Sb_2_O_3_ dielectric substrate exhibit negligible hysteresis [10]. However, the dielectric constant of Sb_2_O_3_ is also only 11.5, there is still a lot of room for improvement.

As a member of transition metal oxides (TMO), α-MoO_3_ is widely used in electronics, sensors, energy storage and other fields [11]. More importantly, α-MoO_3_ has high dielectric constant. It has been shown that α-MoO_3_ has the relative dielectric constant as high as 35, near 10 times that of SiO_2_ [12]. In addition, the unique vdW characteristics can be easily integrated with other 2D vdW semiconductors, promising applications in future 2D electronics and optoelectronics [13,14].

Here, we constructed the α-MoO_3_ flake support MoS_2_ field effect transistor. Thanks to high capacitive coupling, we achieve a low-threshold voltage (−1 V), which is lower than hBN (−2 V) and Sb_2_O_3_ (−7 V), and has greater electron mobility (85 cm^2^ V^−1^ s^−1^) compared to the SiO_2_ dielectric layer (41 cm^2^ V^−1^ s^−1^). In addition, due to the good response of MoS_2_ to visible light, combined with the excellent performance of the MoO_3_ gate dielectric layer, we can obtain photodetectors with higher responsivity (5 A/W).

## 2. Results and Discussion

### 2.1. Structural Characteristics of α-MoO_3_ Flakes

α-MoO_3_ is a wide-bandgap material (3 ev), has a large work function (6 ev), and is an orthorhombic crystal structure [15,16,17]. As shown in Figure 1a, it consists of double layers of MoO_6_ octahedrons of approximately 1.4 nm thickness. In each layer, the octahedrons are arranged by sharing angles in the directions [001] and [100], and the different layers are arranged in the direction [010] by van der Waals forces. The preparation of uniform large-size MoO_3_ nanosheets is the key to the application of the gate dielectric, so several α-MoO_3_ nanosheets were grown using rapid-cooling chemical vapor deposition (CVD) (growth details are shown in the experimental section and Appendix A) [18]. Figure 1b shows the optical image of the prepared MoO_3_ nanosheet. It can be seen that MoO_3_ nanosheet has a large area, with micron level length and width, regular shape and smooth surface. As show in Figure 1c, the layered structure of MoO_3_ is well illustrated by the multi-thickness steps at the edge in atomic force microscopy (AFM) image. XRD characterization revealed that the lattice has a series of sharp peaks at 13.7°, 26.6° and 39.8°, which correspond to the (020), (040), and (060) planes of α-MoO_3_, indicating the high quality of MoO_3_ single crystal (Figure 1d).

### 2.2. Device Performance

Good insulation is the key to gate dielectric applications, therefore, we first studied the insulation of MoO_3_ before preparing heterojunction devices. As shown in Appendix A, when the source–drain voltage (*V_ds_*) is 2 V, the MoO_3_ FET device presents a high resistance state of 10^−13^ A and remains stable under the action of gate voltage, suggesting the good insulation of MoO_3_. Furthermore, we constructed a graphene/MoO_3_/MoS_2_ field effect devices by the dry method of location transfer where graphene, MoO_3_ and MoS_2_ are the gate electrode, gate dielectric layer and channel layer, respectively (details are shown in the experimental section and Appendix A). An optical image of the prefabricated device is shown in Figure 2b. The AFM in Figure 2c shows that the corresponding thickness of MoS_2_, MoO_3_ and Gr are 50 nm, 90 nm and 50 nm, respectively. Raman spectroscopy is further used to confirm heterojunction composition. In the Raman spectrum of graphene, the G peak near 1580 cm^−1^ can correspond to the c–c σ bond of carbon atoms, reflecting the in-plane vibration of carbon atoms, while the G’ peak near 2700 cm^−1^ corresponds to carbon atom π-bond in the z direction, reflecting graphene-layered connections [19,20]. In the Raman spectrum of MoS_2_, the $E_{2g}^{1}$ peak located near 386 cm^−1^ is caused by the opposite vibrations of the two S atoms relative to the Mo atom, while the A_1g_ peak located near 408 cm^−1^ is due to the S atoms in opposite directions out-of-plane vibration [21]. In the Raman spectrum of MoO_3_, there is a strong Raman peak at 820 cm^−1^, which corresponds to the O_3_-Mo-O_3_ stretching vibration in the [100] direction [22]. Due to the special structure of MoO_3_, there may be oxygen vacancies inside, which will affect the electron mobility, band gap, etc. [23]. Thus, we calculated the exact MoO_3_ stoichiometry by analyzing the Raman spectrum. From the Raman spectrum of MoO_3_ in Figure 2d, the ratio of I_512_/I_820_ is approximately 3.997, and finally, we calculated the ratio of O/Mo to be approximately 2.998, which indicates that there are very few oxygen vacancies in the MoO_3_ nanosheets we used. In the Gr/MoO_3_/MoS_2_ heterojunction region, Raman signals corresponding to different materials can be obtained simultaneously, which proves the composition of the heterojunction.

MoS_2_ FETs using MoO_3_ nanosheets as the gate dielectric show interesting device performance. Figure 3b shows the transfer curves of MoS_2_ FETs with different gate dielectric layers. Compared with SiO_2_ as the gate dielectric layer, it can be found that when the MoO_3_ nanosheet is used as the gate dielectric layer, MoS_2_ channel conductance increases with the increase in gate voltage, and obvious on–off current can be observed, showing obvious n-type modulation characteristics, which is caused by the natural sulfur vacancy of MoS_2_ [24]. Additionally, due to the low gate current leakage of ~10^−11^ A, this FET exhibits an equally high on/off ratio (~10^5^). However, the MoS_2_ transistor using SiO_2_ as the gate dielectric layer cannot achieve obvious on–off state current within the same gate voltage range. By further increasing the gate voltage, we still observe that MoS_2_ channel conductance increases with increasing V_gs_, because a larger V_gs_ may gradually open the whole channel and increase the conductivity of the whole channel. At the same time, it requires a more negative V_gs_ to close the channel, and a threshold voltage of approximately −9 V can be observed (Figure 3a). This indicates that MoS_2_/MoO_3_ FET has lower starting voltage and lower power consumption. In addition, the hysteresis of a transfer curve is usually used to study the charge trap state at the interface. The change of threshold voltage $\Delta V_{th}$ in the double-scan transmission characteristic curve is usually affected by the charge ΔQ of the trap state density. It can be found that, with vdW dielectric, ultra-small hysteresis can be observed in the double scan, and the MoS_2_/SiO_2_ FET hysteresis window is quite big. According to
$$\Delta Q=\Delta V_{th}\times C$$
where C and ΔQ represent the gate capacitance and capture charge, respectively, the trap state density can be obtained as 9.8 × 10^10^ cm^−2^, while the trap state density of SiO_2_ is 1.4 × 10^12^ cm^−2^, indicating that MoO_3_ is within low effective trap state density. By processing transfer curve data (Figure 3b), we obtained that the subthreshold swing (SS) of molybdenum disulfide with MoO_3_ as the gate dielectric layer is basically stable at approximately 400 mV/dec. The gate control characteristics of the channel conductance and the ohmic contact between the electrode and MoS_2_ can be further verified from the output curve in Figure 3c. In order to clearly compare the effect of dielectric effects (MoO_3_ and SiO_2_) on FET mobility, we plot together their electron mobility, which is modulated by the gate voltage (Figure 3d). The electron mobility can be calculated by the following formula:(1)$$u=\frac{dI_{ds}}{dV_{g}}\times \frac{L}{W}\times \frac{1}{C\times V_{d}}$$
where L and W are the channel length and width, respectively, C is the capacitance, and C can be calculated as 3.27 × 10^−7^ F/cm^2^ by the following formula:(2)$$C=\frac{\varepsilon_{0}\varepsilon_{r}}{d}$$
where $\varepsilon_{0}$, $\varepsilon_{r}$ and d are vacuum permittivity, relative permittivity and dielectric thickness of dielectric layers, respectively.

It was evident that MoS_2_ on MoO_3_ substrate showed approximately twice the mobility (85 cm^2^ V^−1^ S^−1^) compared with that on the SiO_2_ substrate (41 cm^2^ V^−1^ S^−1^).

Finally, we statistically compared the performance of this device with other MoS_2_ devices with difference gate dielectric layers (Table 1), showing that the MoO_3_ dielectric layer can still have a larger current switching ratio under the small threshold voltage, and its requirements on material thickness is also low.

### 2.3. Device Mechanism Exploration

In order to further understand the carrier transport characteristics of heterojunction devices, Kelvin probe force microscopy (KPFM) was used to obtain the function of the surface potential position along the channel of the sample. Figure 4a is a schematic diagram of KPFM experiments on heterogeneous junction devices under different working conditions. We fixed the source–drain voltage V_ds_ = 0 V and studied the functional relationship between the surface potential and back-gate voltage (V_gs_). As shown in Figure 4b, since both ends of the source and drain are grounded, the corresponding surface potential is 0 V, while the channel has an obvious potential difference. When the gate voltage is between −10 V and −2 V, there is no obvious change in the surface potential difference. When the gate voltage rises to within the range of −2 V–0 V, the surface potential of the channel gradually rises, and then it exceeds the source–drain surface potential, and the potential difference is 0.5 V; the gate voltage continues to rise, the surface potential continues to rise, but the upward trend has weakened. Through the change of surface potential difference and the corresponding transfer curve, it can be found that when the gate voltage is −10 V, the surface potential is the smallest, which means that the contact barrier is low, electrons are prone to transition at the interface, and the source–drain current has a large upward trend. As the gate voltage increases, the surface potential difference rises, so the contact potential barrier is raised rapidly, and it is difficult for electrons to transition at the interface, and finally the rising trend of the source–drain current is weakened.

### 2.4. Device Specific Performance Parameters

Furthermore, the photoelectric performance of the device is verified. We irradiated the entire surface of the device with a 405 nm laser, adjusted the different laser powers and tested whether the relationship between the source–drain current and the gate voltage would be changed compared with the dark state (Figure 5a). As can be seen from Figure 5a, the photoelectric response of the device is more obvious under the action of negative gate pressure. According to the following formula:$$R=\frac{I_{light}-I_{dark}}{PA}$$
where R is photoresponsivity, P is the optical power density, A is the effective area of the device, $I_{light}$ and $I_{dark}$ denote the photocurrent and dark current, respectively.

The highest responsivity of the device can be obtained as the power is 1 mW/cm^2^ in Figure 5b. Figure 5c summarizes the functional relationship between the responsivity of MoS_2_ on different substrates and the gate voltage. Since the difference between the photocurrent and dark current of MoS_2_/MoO_3_ is larger than that of MoS_2_/SiO_2_, by comparison, it is found that the responsivity of MoS_2_/MoO_3_ is significantly higher than that of MoS_2_/SiO_2_ under the same optical power, indicating that MoS_2_ on MoO_3_ substrate shows superior photoelectric performance.

## 3. Conclusions

We demonstrate that MoS_2_/MoO_3_ can effectively reduce the device operating voltage. The device exhibits a high on–off ratio (10^5^), low SS (400 mV/dec), high electron mobility (85 cm^2^ V^−1^ s^−^^1^), low lag and high stability due to fewer surface defects. In addition to this, we obtained devices with higher electron mobility and a lower density of defect states, which favors smaller hysteresis voltage. Furthermore, the device is capable of significant response to visible light and the most responsivity is near 5 A/W. In general, this study proves that high-κ van der Waals MoO_3_ can be effectively applied to the gate dielectric layer to improve the photoelectric performance of the device.

## 4. Experimental Section

**MoO_3_ Single Crystal Growth:** We used the vapor deposition method, placed approximately 0.1 g of MoO_3_ powder in the center of the ark, and then placed the ark in the center of the tube furnace, continued to pass in the gas Ar at a rate of 100 sccm, and connected the gas outlet to the atmosphere. The heating started from room temperature (25 °C), 25 min to 780 °C, keeping warm for 1 min, then the heating tube was quickly removed, and the tube furnace was allowed to cool down rapidly, during which MoO_3_ nanosheets could be seen flying out and depositing on the silicon wafer approximately 10 cm away from the powder on the side near the gas outlet. This resulted in large-area MoO_3_ nanosheets.

**Gr/MoO_3_/MoS_2_ Heterostructure Memory Devices Fabrication:** We used the mechanical exfoliation method, used the layered structure of 2D materials, used tape to obtain thin-layer materials, and transferred the thin-layer materials to the same silicon wafer through PDMS. Appendix A shows the transfer schematic diagram, and displays the transfer and stacking of Gr, MoO_3_ and MoS_2_ from left to right, respectively. Then, we used the flocculated crystals generated in the tube furnace during the growth of MoO_3_ nanosheets to cover the surface of the transistor part with the probes. The surface of the transistor is covered with gold by evaporation method, and the channel of the transistor can be obtained by removing the crystal. Finally, the device is obtained by drawing electrodes with a probe.

**Device Characterization:** AFM (NT-MDT NTEGRA) was used to characterize the change of surface morphology. The photoluminescence spectrum was measured using confocal Raman microscope (Horiba LabRAM HR Evolution Inc., Palaiseau, France). Semiconductor device parameter analyzer (FS-Pro, Primarius Electronic Technology, Beijing, China) was employed to characterize the electrical properties of the devices. Semiconductor lasers with a wavelength of 405 nm were used to study the photoresponse properties of the devices.

## Figures and Tables

**Figure 1 materials-15-05859-f001:**
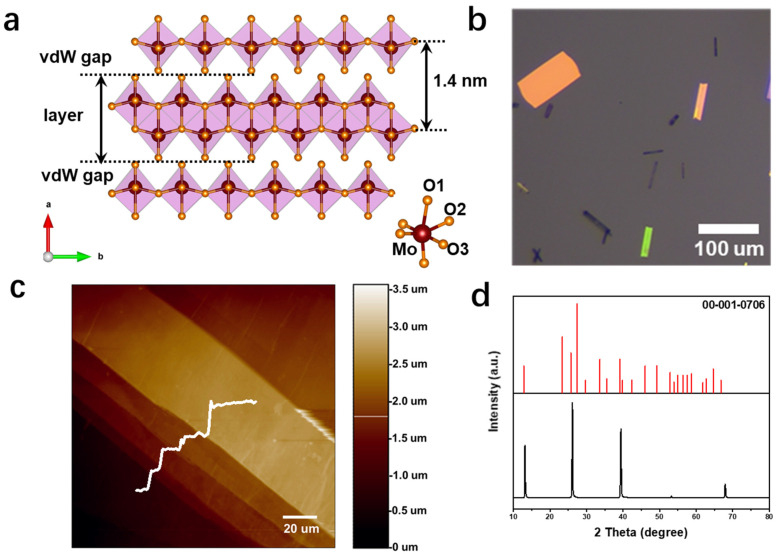
(**a**) MoO_3_ lattice structure diagram. (**b**) Optical microscope image of MoO_3_ nanosheets grown by vapor deposition method. (**c**) AFM surface topography of MoO_3_ nanosheets. (**d**) XRD spectrum of MoO_3_ nanosheets.

**Figure 2 materials-15-05859-f002:**
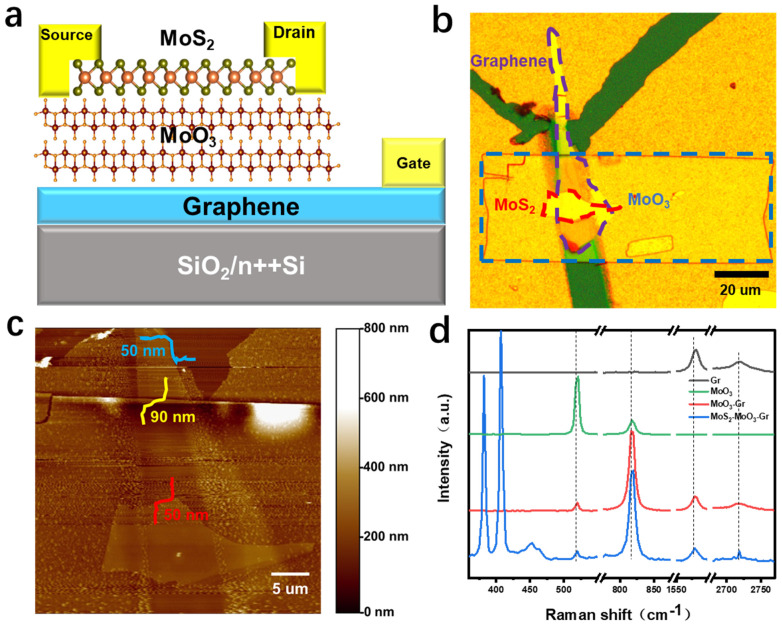
(**a**) Schematic diagram of Gr/MoO_3_/MoS_2_ heterostructure. (**b**) Optical image of the device under optical microscope. (**c**) AFM surface topography of the device. (**d**) Raman spectra at different positions of the device.

**Figure 3 materials-15-05859-f003:**
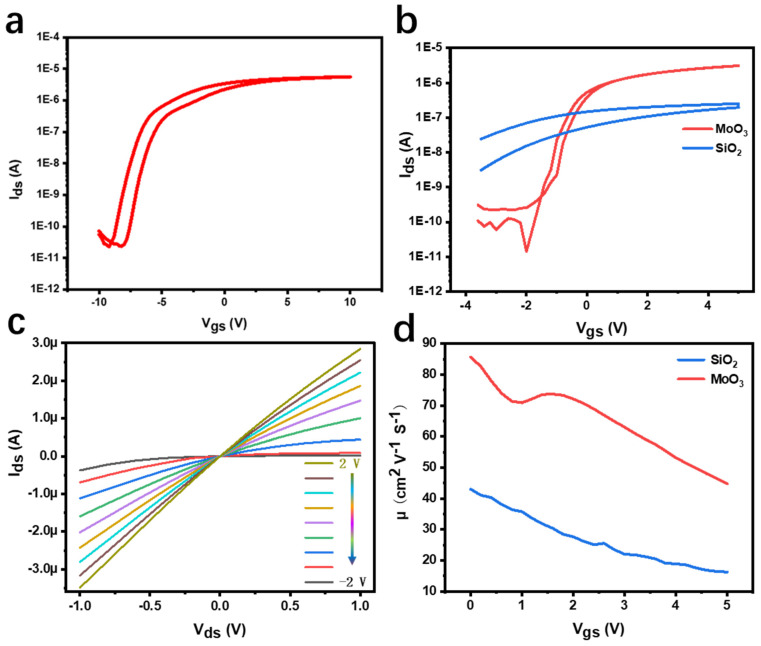
(**a**) Transfer curves of the gate of SiO_2_ under more gate voltage range. (**b**) Comparison of transfer curves of different substrates under the same gate voltage range. (**c**) Output curves of the device, the interval is 0.5 V. (**d**) The relationship between the mobility of different substrates and gate voltage curve.

**Figure 4 materials-15-05859-f004:**
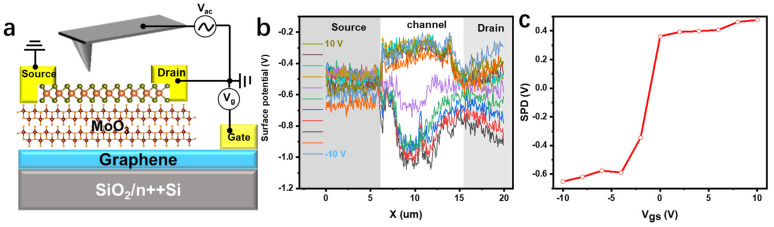
(**a**) Schematic diagram of KPFM test structure. (**b**) Surface potential diagram at different gate voltages, the interval is 2 V. (**c**) Summary diagram of contact potential difference between electrode and MoS_2_ under different gate voltages.

**Figure 5 materials-15-05859-f005:**
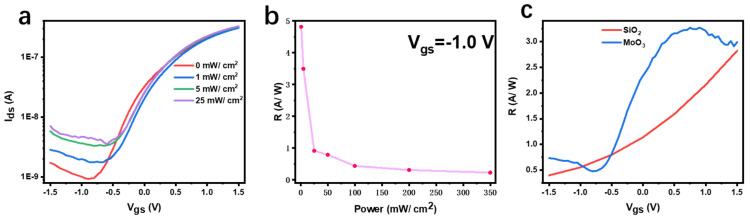
(**a**) Transfer curves of a 405 nm laser with a different light power in MoO_3_ medium. (**b**) The relationship between responsivity and optical power at the same gate voltage in MoO_3_ medium. (**c**) The relationship between the responsivity and gate voltage of the two media under the same light power (50 mW/cm^2^).

**Table 1 materials-15-05859-t001:** Device data for different gate dielectric layers.

Gate Material with MoS_2_	hBN [25]	ZrO_2_ [26]	Al_2_O_3_ [8]	HfO_2_ [7]	Sb_2_O_3_ [10]	MoO_3_ (This Work)
Thickness (nm)	7	27	100	5	40	90
Threshold voltage (V)	−2	−6	−1	−1	−7	−1
Current switching ratio	10^3^	10^5^	10^6^	10^6^	10^8^	10^5^
Electron mobility (cm^2^ V^−1^ s^−1^)	45	6.9~11.5	200	8	70~90	85
Trap states density (cm^−2^)	1.9 × 10^11^	3 × 10^12^	-	1 × 10^13^	6.9 × 10^9^	9.8 × 10^10^
SS (mV/dec)	57	276	-	110~120	-	400

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
