# Peer review of "High-κ van der Waals Oxide MoO3 as Efficient Gate Dielectric for MoS2 Field-Effect Transistors"

_materials, 2022, doi:10.3390/ma15175859_

Round 1
Reviewer 1 Report
Dear Authors,
I think "High-κ van der Waals oxide MoO3 as efficient gate dielectric for 2D field-effect transistors" is a well written manuscript.
The only thing you should improve is the introduction. I think a separate section describing also the setback of working with such delicate materials should be insert in the text. For example, oxygen vacancies, even a small quantity may change the properties of MoO3 as dielectric gating. A comparison with a very recent study (P-type Ohmic contact to monolayer WSe2 field-effect transistors using high electron affinity amorphous MoO3, 10.48550/arXiv.2206.11096) should be also done.
Please also add a small part in the analysis. You should calculate the exact MoO3 stoichiometry (i.e. the presence of oxygen vacancies) from Raman spectrum as showed in this recent article: "Metallic Interface Induced Ionic Redistribution within Amorphous MoO3 Films, DOI: 10.1002/admi.202200453"
This is a very powerful analysis tool and in this way you can be more accurate and the article will gain additional precise information.
Reviewer 2 Report
Please refer to the attached file for more details.

Reviewer 3 Report
The authors have done excellent work on a topic that is very important for next node semiconductor devices. A few comments:
1. Please fix grammatical errors
2. Please explain the fabrication of the transistor in greater detail. The current level of detail is not sufficient to replicate your process.
3. With monocrystalline MoO3, do you anticipate needing multiple MoO3 samples in a scaled up device? If so, please comment on how you will prevent grain boundaries from behaving as a leakage path. If not, please explain how your current experimental design can be scaled up.
4. Please include Raman spectra specific to your device in your manuscript. While G peak is important, it is also essential to study G/D ratio and to see if 2D peak is present – indicating graphene. This is crucial in understanding if the graphene quality is affected because of processing.
5. Section 2.3: Please comment on thermal stability of the MoO3 transistor. Since source drain current gradually weakens please explain the mechanism. Does diffusion of MoO3 into underlying layers hint at root cause? You can verify this by examining the crystal structure of MoO3 before and after measurement.
6. Section 2.4: Please explain the mechanism behind better performance of MoS2/MoO3 vs MoS2/SiO2.
Round 2
Reviewer 2 Report
Authors answered all suggestions and comments, I have only one remark:
Table 1 and its corresponding text should be moved to the results and discussion section (not in the conclusion section) and title (of table 1) should be placed at the top of the table.
